# Graft Polymerization of Acrylamide in an Aqueous Dispersion of Collagen in the Presence of Tributylborane

**DOI:** 10.3390/polym14224900

**Published:** 2022-11-13

**Authors:** Yulia L. Kuznetsova, Karina S. Sustaeva, Alexander V. Mitin, Evgeniy A. Zakharychev, Marfa N. Egorikhina, Victoria O. Chasova, Ekaterina A. Farafontova, Irina I. Kobyakova, Lyudmila L. Semenycheva

**Affiliations:** 1Federal State Budgetary Educational Institution of Higher Education, Privolzhsky Research Medical University of the Ministry of Health of the Russian Federation, 603005 Nizhny Novgorod, Russia; 2Department of Organic Chemistry, Faculty of Chemistry, National Research Lobachevsky State University of Nizhny Novgorod, 23, Gagarin Ave., 603022 Nizhny Novgorod, Russia

**Keywords:** collagen, polyacrylamide, tributylborane, hybrid copolymer, cytotoxicity

## Abstract

Graft copolymers of collagen and polyacrylamide (PAA) were synthesized in a suspension of acetic acid dispersion of fish collagen and acrylamide (AA) in the presence of tributylborane (TBB). The characteristics of the copolymers were determined using infrared spectroscopy and gel permeation chromatography (GPC). Differences in synthesis temperature between 25 and 60 °C had no significant effect on either proportion of graft polyacrylamide generated or its molecular weight. However, photomicrographs taken with the aid of a scanning electron microscope showed a breakdown of the fibrillar structure of the collagen within the copolymer at synthesis temperatures greater than 25 °C. The mechanical properties of the films and the cytotoxicity of the obtained copolymer samples were studied. The sample of a hybrid copolymer of collagen and PAA obtained at 60 °C has stronger mechanical properties compared to other tested samples. Its low cytotoxicity, when the monomer is removed, makes materials based on it promising in scaffold technologies.

## 1. Introduction

Organoboron compounds occupy a special place both in the study of the polymerization of a wide range of monomers and in the production of finished goods. This results from the ability of alkylboranes to participate in all elementary stages of radical polymerization [1,2]. The alkylborane–oxygen system is the most promising [3,4,5,6,7,8,9,10,11]. On the one hand, it is a low-temperature radical initiator, suitable for use in graft polymerization, and on the other, a reversible inhibition agent. The combination of these properties determines the system’s potentially wide application in (co)polymer synthesis and in macromolecular design. It should be noted that, frequently, alkylboranes themselves are not used directly to solve the relevant tasks, but instead, are substituted with oxidation-stable amine complexes that release alkylboranes during the polymerization. Papers [5,6] report on the pseudo vivo (reversible) radical polymerization of alkyl(meth)acrylates initiated by the binary trialkylborane–oxygen system. A method exists of modifying the surface of polypropylene to change its properties through graft polymerization of alkyl acrylates [8] or maleic aldehyde [9] using the alkylborane–oxygen system. Papers [10,11] report on a room temperature polymerization method under the action of the alkylborane amine complex, using acrylic monomers of methyl acrylate (MA) and copolymers of MA, and AA along with linear copolymers of polydimethylsiloxane with isopropylacrylamide [11], vinylpyrrolidone [10], etc. The alkylborane–oxygen system is already known as a low-temperature initiator for the radical polymerization of vinyl monomers for bonding thermoplastics and materials with low surface energy [12]. The approach of using trialkylborane amine complexes with an acrylate base originally attracted attention in adhesive development because of the speed of graft polymerization on the substrate surface, being a rapid curing material with a unique ability to promote adhesion to plastics with low surface energy (polypropylene, polyethylene, Teflon) due to the involvement of an extremely active boron centered radical [13]. Radical copolymerization using trialkylboranes is a method that can produce collagen-based hybrid materials, as described in recent literature [14]. TBB is used as a radical copolymerization initiator to obtain hybrid copolymers of methyl methacrylate and collagen [15], methyl methacrylate and gelatin [16], AA and gelatin [17], and butyl acrylate and collagen [18]. In the case of collagen, TBB enables polymerization even at 25 °C, avoiding collagen denaturation and thus maintaining its structure [15]. It should be noted that polymerization in the presence of an alkylborane–oxygen initiator system requires no heating or UV irradiation, this being a particular advantage for the encapsulation of heat-sensitive hydrophilic active substances into silicone copolymer particles [10,11]. This advantage also allows the use of the alkylborane oxygen system in the production of materials for 3D printing and in the synthesis of hybrid products incorporating natural polymers [15,16,17,18]. Given the uniqueness of organoboranes as important reagents in radical (co)polymerization, research in this field remains relevant and makes it possible for new aspects of their capabilities to be discovered. Such versatile participation of organoboranes in radical, including graft, polymerization contributed to the choice of these compounds for the synthesis of scaffold precursors based on fish collagen. The prevalence of fish collagen compared to animal collagen is currently very high [19,20,21]. This is due to the following factors: 96% identity to human collagen; hypoallergenic and transdermal properties; inertia towards viruses, as well as the absence of religious restrictions. The use of fish collagen in the production of dressings and coatings [20,21,22,23,24,25,26,27] is due to its biocompatibility, biodegradability, and low antigenicity.

Scaffold technologies require the creation of materials with the properties listed above, as well as certain performance characteristics. The solution to this problem is the creation of hybrid materials based on collagen and synthetic polymers [28,29,30,31,32,33,34,35,36,37,38] with the participation of specialists from various specialties: medicine, biology, physics, chemistry, etc. Since PAA is widely known as a component of hybrid copolymers used in medicine as hydrogels [39], biodegradable drug carriers [40], and also has a porous structure, which makes it a promising component in the creation of scaffolds, we considered PAA as a synthetic component of a hybrid copolymer.

The current research is aimed at investigating AA graft polymerization in aqueous dispersions of collagen in the presence of TBB as the initiating agent at various temperatures, the determination of molecular weight characteristics, structure, mechanical properties, and cytotoxicity of the obtained copolymers, and the evaluation of the prospects for using the obtained copolymers in scaffold technologies.

## 2. Materials and Methods

### 2.1. Materials

An acetic acid dispersion of collagen was obtained from cod skin using a patented technique [41]. Collagen characteristics: Mn = 170 kDa, Mw = 210 kDa, PDI = 1.20. A 1% collagen concentration in the dispersion was obtained by diluting with 3% CH_3_COOH solution. The acetic acid dispersion was purified by recrystallization from benzene, chloroform was dried with heat-treated calcium chloride, then distilled and stored in a dark vessel [42].

The TBB was synthesized using the technique described in [17]. Mg shavings (19.46 g, 0.8 mol) were placed into a 2 L three-necked flask equipped with a mechanical stirrer and reflux condenser; the mixture was heated and cooled in an argon atmosphere. Then, BF3-Et2O (28.2 g, 0.2 mol), iodine crystals, and anhydrous diethyl ether (200 mL) were added to the reaction flask still under argon. The reaction was then initiated by adding 9.4 mL of 1-butyl bromide (1) dropwise while stirring the reaction mixture, and residue 1 (73.8 g, 0.6 mol) dissolved in ether (100 mL) was slowly added over the space of 1 h, the ether being gently boiled using a reflux condenser. Stirring was continued for another 1.5 h; after completion of the addition of 1, water (3.6 mL) saturated with NH_4_Cl was added. The reaction mixture was allowed to stand until the clear ether supernatant could be decanted into a distillation flask. The ether was then distilled off in an argon flow and the residual TBB was distilled off in a vacuum. ^11^B NMR (CDCl_3_, δ, m.d., J/Hz): 86.7, J = 128 MHz.

### 2.2. Polymerization

An amount of 30 mL samples of the 1% collagen acetic acid dispersion were placed into two-necked flasks filled with argon and heated in a water bath to 25, 45, or 60 °C, respectively. For each of these flasks, 0.08 g of TBB was placed in an ampoule, the mixture was degassed by freezing and thawing repeatedly under vacuum, then the ampoule was filled with argon. The contents of the ampoule containing the TBB were then poured into one of the argon-filled reaction flasks and incubated for 30 min at the appropriate temperature, with constant stirring. Then, a degassed solution containing 0.3 g AA in 3 mL water was added to the reaction flask. The reaction mixture was incubated for another 3 h.

### 2.3. Chromatography–Mass Spectrometry

The collagen was freeze-dried, placed in an ampoule, and vacuumized; then, a heptane solution of TBB was added, the mixture was incubated for 30 min, and the ampoule was filled with argon. The gas phase was sampled and analyzed on a QP-2010 (Shimadzu) quadrupole chromato-mass spectrometer, using 70 eV electron impact ionization. Separation was performed at 40 °C on a DB-1 column (30 m, 0.25 mm, 0.25 µm). The evaporator, transition line, and ion source temperatures were all 200 °C. Helium was used as the carrier gas, with a 50 kPa column inlet pressure, a 1:20 sample separation, and 0.5 mL sample injection volume. The chromatogram was recorded in positive ion mode, using a total ion current in the 40–200 Da range.

### 2.4. Fourier Transform Infrared Spectroscopy

For this, copolymer films were prepared on KBr plates. The IR absorption spectra were recorded on an “IRPrestige-21” FTIR-spectrophotometer (Shimadzu, Kyoto, Japan). 

### 2.5. Gel Permeation Chromatography

The collagen and PAA copolymers’ water dispersion was analyzed on an LC-20 HPLC system (Shimadzu, Japan) with an ELSD-LT II low-temperature light-scattering detector. Measurements were performed in the following conditions: the column was a Tosoh Bioscience TSKgel G3000SWxl (30.0 cm L, 7.8 mm i.d., 5.0 µm pore size), at a column temperature of 30 °C, the mobile phase was 0.5 M acetic acid in water, and the injection volume was 20 µL, with a flow rate of 0.8 mL/min. Calibration was performed using narrow dispersion dextran with a molecular weight (MW) range of 1–410,000 Da (Fluca). SEC data processing was performed with LC-Solutions-GPC software. 

### 2.6. Determination and Removal of the Unreacted Monomer

Unreacted AA was measured by bromination according to the Knopp method. Bromine was generated by the reaction of a bromide–bromate solution (5.568 g KBrO_3_, 40 g KBr in 1 L of water) and hydrochloric acid. An aqueous dispersion of copolymer ~2 g in 100 mL of water was placed into a flask and 25 mL of the bromide–bromate solution and 10 mL 10% hydrochloric acid were added, the mixture was stirred and left in a dark place for 2.5 h. Then, 15 mL of 10% KI was added, and the released iodine was titrated with a 0.1 N solution of Na_2_S_2_O_3_. A control was conducted with distilled water. The AA concentration was determined by: 

*X*% = ((*a − b*) × *M*)/(*200 × m*), where a and b are the volumes (in mL) of Na_2_S_2_O_3_ used for the probe and control titrations, respectively; M is the molecular weight of the AA, g/mole; and m is the mass of the copolymer probe, g. Both the mass and percentage of the unreacted AA were calculated.

To remove the unreacted monomer the freeze-dried, weighed samples of copolymer were extracted with chloroform in a Soxhlet extractor for 40 h. Then, they were dried to reach stationary mass, weighed and the unreacted AA was calculated as described above.

### 2.7. Scanning Electron Microscopy

Nanometer resolution images of the surfaces of the collagen and copolymer films were taken with a JEOL JSM-IT300LV scanning electron microscope in low vacuum at 4.0 nm resolution, using a 30 kV accelerating voltage and 5 to 2000× magnifications to provide prints 10 cm × 12 cm in size. The samples were obtained by removing small pieces of the freeze-dried polymer films from the substrate.

### 2.8. Kinematic Viscosity Determination

The flow times of a 1% collagen solution and of further dilutions of solutions of the AA-collagen copolymers synthesized at 25, 45, and 60 °C initially diluted to 1% were determined using a d = 0.56 PZH-2 viscometer until the results converged. For the measurements, the viscometer was filled and placed in a thermostat heated to 25 °C. The kinematic viscosity was determined according to the formula:

V = $\frac{g}{9807}\;$× T × K, where:

K–viscometer constant (K = 0.011797);

V–fluid kinematic viscosity, mm^2^/sec; 

T–expiration time of the fluid, sec.

### 2.9. Measurement of the Mechanical Properties of Films Based on the Synthesized (co)polymers

The mechanical properties of films obtained by irrigation from the synthesized (co)polymer solutions were measured using an AG-Xplus 0.5 universal testing machine (Shimadzu, Japan). For testing, 5 mm × 40 mm rectangular samples of the 20 to 80 µm thick films were cut. The tensile test was carried out at a speed of 1 mm/min. The maximum stress (strength) and elongation at breakage were recorded. The average value of 5 tested samples was taken as the measurement result.

### 2.10. Cytotoxicity Assessment Using MTT Assay

#### 2.10.1. Testing with Cultures

Human dermal fibroblasts (HDFs) were used as MTT assay samples. Active, morphologically homogeneous 5–6 passage cultures with cells that adhered well to the plastic were used. The culture cells’ immunophenotype corresponded to that of mesenchymal cells and the culture viability was 95 to 98%. The culture of HDFs used in this study had previously been tested for sterility and infection.

#### 2.10.2. MTT Assay

The MTT test is a colorimetric quantitative test used to measure cellular metabolic activity and viability. The method is based on the reaction of 3-(4,5-dimethylthiazol-2-yl)-2,5-tetrazolium bromide (MTT) being reduced to purple formazan inside living cells. The extent of MTT recovery depends on the metabolic activity of NADPH-dependent oxidoreductase enzymes; accordingly, living and actively dividing HDFs show a high degree of MTT recovery, while HDFs that are toxically damaged (low metabolic activity), or dead show a low degree of MTT recovery. Dimethyl sulfoxide (DMSO) is added as a solvent to produce colored solutions as it reduces the intracellular formazan crystals. The color intensity of the test samples after the addition of DMSO was quantitatively measured at a wavelength of 540 nm using a flatbed reader.

The samples tested had the following characteristics: 

Sample I was an aqueous solution of acetic acid containing a dispersion of collagen and PAA copolymer resulting from synthesis at 60 °C. The viscous, turbid-white, translucent liquid, with an acidic pH, was neutralized to pH = 7.3 before examination. 

Sample II was a freeze-dried copolymer sponge of collagen–PAA synthesized at 60 °C, with no further treatment. It was white, brittle, and paper-like. It dissolved completely and had a neutral pH. 

Sample III was a freeze-dried copolymer sponge of collagen–PAA synthesized at 60 °C and then washed with chloroform in a Soxhlet extractor. It was white and brittle. 

Samples II and III were placed in Petri dishes (60 mm diameter) filled with DMEM/F12 medium containing 1% antibiotics (penicillin-streptomycin 100x, PanEco) and 2% fetal calf serum, and were incubated for 1 day at 37 °C and 5% CO_2_. After 24 h, complete dissolution of the samples was observed, and the extract obtained was used for the study. Sample I was a suspension (hereafter—“extract”). For the MTT assay, the extracts obtained were diluted and used in the following proportions: Control 0:1: Extract 1:0; with further dilutions 1:1; 1:2; 1:4, and 1:8 of extract: medium, respectively.

As a preliminary, one day before the MTT assay, HDFs at a concentration of 1 × 10^4^ cells per square centimeter (well size 0.5 cm^2^) were inoculated onto 96-well flatbeds in DMEM/F12 medium with 1% antibiotics (penicillin-streptomycin 100x, PanEco) and 10% fetal calf serum, 200 µL suspension per well. The flatbed with HDFs was then placed in an incubator at 37 °C and 5% CO_2_.

After 1 day of cultivation, the growth medium in the flatbed wells was removed, discarded, and replaced with either the control medium or the above extract dilutions. The flatbeds were replaced in the incubator for 3 days. After 72 h, 20 µL of MTT solution was injected into each well and the flatbeds were replaced in the incubator for another 3 h. After 3 h, the liquid was gently removed from all the wells and replaced with 99% DMSO solution, 200 µL per well. The optical density (OD) was measured using a Sunrise analyzer (Austria).

The relative growth intensity (RGI) was calculated according to the formula:$$\mathrm{RGI}(\%)=\frac{\mathrm{mean}\;\mathrm{OD}\;\mathrm{in}\;\mathrm{testing}\;\mathrm{culture}}{\mathrm{mean}\;\mathrm{OD}\;\mathrm{in}\;\mathrm{control}}x100$$

The results were evaluated using a cytotoxicity grading scale (Table 1).

The 0–1–rank indicates the absence of cytotoxicity, 5 rank–maximum cytotoxicity.

## 3. Discussion of the Results

Collagen is a common, natural component used for biodegradable hybrid copolymers with synthetic fragments; but, at elevated temperatures, collagen undergoes denaturation, gradually transforming into gelatin, so it is important to study the effect of temperature on the structure of the copolymers in which it has been used. For collagen and polymethyl methacrylate (PMMA) copolymers obtained in the presence of TBB within the 25–60 °C temperature range, it was found that the collagen structure was only preserved at a 25 °C synthesis temperature [15]. Thus, although the molecular weight characteristics and composition of the copolymers have similar values, their properties depend on the synthesis temperature, e.g., the light transmission is higher for PMMA copolymer films and collagen films synthesized at 60 °C, where the collagen structure is already disrupted. It has previously been shown [43] that there are only insignificant differences between the properties of cod gelatin and cod collagen (CC) subjected to enzymatic protein hydrolysis, and in their functional properties in hybrid hydrogel scaffolds in combination with fibrinogen. This suggests that it is potentially valuable to use CC in hybrid compositions and at temperatures above room temperature. Furthermore, the selected temperature profile will allow researchers to vary some properties of the target product. We synthesized collagen and PAA copolymers at the previously suggested temperatures of 25, 45, and 60 °C [15]. The process was run under conditions minimizing the formation of active radicals through tributylboron oxidation. The component ratios of the polymerizing composition were chosen based on the results of [17] with copolymers of gelatin and PAA synthesized at different concentrations of the TBB initiator: the TBB was first injected into an acetic acid dispersion of collagen in an inert argon atmosphere (Figure 1). As a result, boroxyl fragments formed at the protein surface due to a reaction with fragments of hydroxyproline units, promoting controlled radical graft polymerization of the synthetic monomer via a reversible inhibition mechanism (Figure 2):

At the end of the copolymerization, a clear colorless dispersion was obtained, which was examined by infrared spectroscopy (Figure 1) and gel permeation chromatography (GPS) (Figure 2).

In the infrared spectrum of the obtained copolymer dispersions (Figure 1, curves 1, 2, and 3), absorption bands related to those of collagen (Figure 1, curve 4) and PAA (Figure 1, curve 5) can be observed.

According to the IR spectra of the obtained copolymers, the intensities of the bands increase at 3275 cm^−1^ this being related to NH_2_ group valence vibrations ν(N-H), while the absorption bands in the 1640 cm^−1^ region were related to valence vibrations ν(C = O) (amide I), and at 1540 cm^−1^ they were related to planar strain vibrations δ(N-H) (amide II) that occur when the synthesis temperature rises to 60 °C (Figure 1, curve 1). This is due to the freeing of amide groups when hydrogen bonds are broken inside the collagen triple helix on heating. Additionally, as the temperature of synthesis rises, intensity increases in the 1720 cm^−1^ range absorption band, this being related to the carboxyl group, perhaps occurring because of partial hydrolysis of the collagen molecules (Figure 1, curve 1). No noticeable changes corresponding to these were observed at 25 °C or 45 °C.

The dispersion contains a highly polymeric product with a molecular weight (MW) slightly higher than that of collagen, as shown by the GPC analysis (Table 2). The molecular weight distribution (MWD) of the obtained copolymers at 25 (Figure 2, curve 2), 45 (Figure 2, curve 3) and 60 °C (Figure 2, curve 4) show a slight shift to higher MWs compared to the original collagen MWD (Figure 2, curve 1). The trend of copolymer MW as a function of the synthesis temperature is consistent with the kinematic viscosity changes found for these samples (Table 2).

The acetic acid copolymer dispersions obtained after grafting contain unreacted AA amounting to 20.5–30.5% as the initial monomer (Table 3), its quantity being greater at higher synthesis temperatures. 

The dispersions were air dried, brought to a constant weight, and, knowing the unreacted monomer content, the PAA content of the highly polymeric products was calculated (Table 3). As can be seen from Table 3, the content of PAA in the product slightly decreases with increasing synthesis temperature. Probably, at 25 °C, a significant part of the process proceeds under the action of the low-temperature initiating system alkylborane–oxygen, where borylated collagen acts as the alkylborane [3,4,5,6]. With an increase in temperature, the proportion of this process decreases, since the rate of oxidation increases, which leads to the formation of low molecular weight products. In this case, polymerization by the mechanism of reversible inhibition becomes predominant (Figure 2), the rate of which is much lower than conventional radical polymerization. All of the above leads, on the one hand, to a decrease in the total conversion of AA and a decrease in MW. 

To purify the copolymers by removing any PAA homopolymer and unreacted monomer, the copolymer dispersions were air-dried to remove water and then placed in a Soxhlet extractor, where they were extracted with chloroform for 40 h, the AA and PAA being soluble in chloroform, while collagen is not. No PAA was detected by IR spectroscopy after the extraction in chloroform, i.e., under these conditions, all the PAA had been grafted onto the collagen, this being consistent with the results of gelatin–PAA copolymer synthesis in the presence of TBB [17]. In addition, extraction of the copolymers with chloroform allowed the unreacted AA to be washed off, as confirmed by Knopp’s method.

The resulting samples were examined using scanning electron microscopy (SEM). The freeze-dried copolymers, together with native collagen and PAA synthesized with a classical radical initiator were used to obtain photomicrographs.

Figure 3 shows significant differences between the copolymer photomicrographs (Figure 3c–e) and those of collagen (Figure 3a) and PAA (Figure 3b). The graft copolymer has denser contours of the collagen matrix due to the grafted synthetic fragments (Figure 3c–e). In particular, the porosity characteristic of the PAA can be observed. The pore size in the copolymers is about 50 µm. The photomicrographs obtained allow us to observe differences in the structure of the collagen fibers in copolymers synthesized at different temperatures. Collagen is characterized by a parallel arrangement of fibers (Figure 3a). With an increase in temperature, the SEM photographs show a noticeable transition from clear parallel collagen fibers (Figure 3a), which are preserved in the copolymer synthesized at 25 °C (Figure 3c), partially preserved in the copolymer synthesized at 45 °C (Figure 3d) and are almost completely absent in the copolymer synthesized at 60 °C (Figure 3e). Thus, collagen fiber integrity is impaired in the copolymer structure with increasing synthesis temperature; this has also been demonstrated for PMMA and collagen copolymers [15].

A wide range of applications for hybrid PAA copolymers occurs in their use as biomedical materials: hydrogels, scaffolds, etc. Pure collagen scaffolds have insufficient mechanical strength, thus limiting their use in tissue engineering. The inclusion of synthetic polymers into the structure provides for a significant increase in the strength of the final material. For all copolymers synthesized at 25, 45, and 60 °C, we measured the maximum force F at which the sample failed and the maximum stress (tensile strength) of sample films compared to the original copolymer constituents (Table 4).

According to the obtained data, grafted copolymers synthesized at 60 °C have stronger tensile properties than their individual constituents, PAA and collagen, respectively. According to the SEM data, such copolymer matrices have a denser structure with more cross-links, hence they are stiffer and can resist additional tensile force. This can be attributed to the fact that when the temperature increases, grafting is no longer carried out mainly on the collagen fibers, but on their denatured analog, i.e., gelatin. The identified differences are related to changes in the gelatin supramolecular structure during collagen denaturation: the collagen molecule is a left-handed helix of three α-chains of amino acid residues of known amino acids around a common axis. The gelatin molecule is a denatured helix with the bonds between the individual α-chains having broken. Unlike collagen fibers, as these parts are no longer bound together by hydrogen bonds, they are therefore more resistant to mechanical stress.

The cytotoxicity parameter is important when such copolymers are used as scaffolds; the cytotoxicity grade can be used to assess the likely extent of cell engraftment on a given material. For our cytotoxicity study using the MTT assay, a collagen and PAA copolymer synthesized at 60°C was selected. This copolymer was characterized by an even pore distribution (Figure 3e) due to the collagen structural breakdown, which also gave it better mechanical properties (Table 4), i.e., it is the most suitable for scaffolding techniques. The data are presented in Table 5 and Figure 4. The dispersion of the collagen and PAA copolymer obtained after synthesis (Sample I) exhibited moderate cytotoxicity—grade 3. It should be emphasized that this toxicity persisted even when the extract was diluted (1:1, 1:2, and 1:4). When examined microscopically, the field of view showed a large number of spherical HDF cells along with solitary flattened cells with regular morphology (Figure 4a). Only with a 1:8 dilution of the Sample I extract could we reduce the cytotoxic effect to cytotoxicity grade 2. However, both spherical cells and flattened HDFs could still be observed in the field of view under a microscope (Figure 4a’). Assuming that Sample I cytotoxicity was related to the presence of unreacted AA (Table 3), a lyophilic-dried copolymer of collagen and PAA synthesized at 60 °C (Sample II) was used for further studies, but, since Sample II also contained unreacted monomer, a lyophilic-dried and chloroform-washed copolymer of collagen and PAA was prepared in a Soxhlet extractor (Sample III) for cytotoxicity studies. The effectiveness of this method for purifying gelatin–PAA copolymer had already been demonstrated in [17]. Indeed, the highest cytotoxicity was observed in the extract and 1:1 dilution of Sample II—cytotoxicity grade 4 (Table 5). When examined microscopically, spherical HDFs with irregular cell membranes were observed, confirming the toxic effects on the culture (Figure 4b). These toxic effects were reduced when the Sample II extract was diluted. A 1:2 dilution demonstrated cytotoxicity grade 2. When examined microscopically, many spindle-shaped cells characteristic of HDFs adhering to plastic were observed, but some of the HDFs still took on a spherical shape. A 1:4 dilution of Sample II demonstrated cytotoxicity grade 1. Visually, the cells looked morphologically correct, with no spherical cells being observed in the field of view, but the cells were not growing as actively as in the control, as evidenced by the OD decrease. A 1:8 dilution (Figure 4b’) was neither visually nor quantitatively different from the control (Figure 4d’).

Sample III showed moderate cytotoxicity with the extract and its 1:1 dilution corresponding to cytotoxicity grade 2. Morphologically correct HDFs like those adhering to plastic were visually observed in these dilutions, but spherical cells were also present in the field of view (Figure 4c). The MTT assay results of dilutions 1:2, 1:4, and 1:8 of Sample III were assessed at cytotoxicity grade 1. When examined microscopically, the condition of the cells at these dilutions (Figure 4c’) corresponded to that of the control (Figure 4d,d’). However, even at these dilutions, cell growth activity was markedly reduced, as evidenced by the optical density values (Table 5). A comparison of the MTT assay results for Sample II and Sample III indicated that the collagen copolymer and PAA cytotoxicity is caused by unreacted AA present as an impurity. Cytotoxicity is reduced by the removal of the unreacted monomer from the copolymer.

## 4. Conclusions

Thus, in the present work, grafted copolymers of PAA and collagen were synthesized in the presence of TBB at different temperatures. The synthesis temperature was found to have no appreciable effect on the composition and molecular weight characteristics of the copolymers. However, collagen denaturation means that the copolymer structure is significantly different when the synthesis temperature is increased. Such a structural change leads to a change in mechanical properties. When cytotoxicity was assessed, the collagen and PAA copolymers synthesized at 60 °C containing an AA impurity (Samples I and II) were found to have pronounced cytotoxicity. This manifestation of cytotoxicity is related to the presence of unreacted monomer in these samples and can be reduced via its removal by extraction with chloroform as in the case of Sample III. The low cytotoxicity of Sample III makes materials based on promising copolymers for further activities aimed at the development of new, artificial cytoskeletons for biomedical applications.

## Data Availability

Not applicable.

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
