# Peer review of "Graft Polymerization of Acrylamide in an Aqueous Dispersion of Collagen in the Presence of Tributylborane"

_polymers, 2022, doi:10.3390/polym14224900_

Round 1
Reviewer 1 Report
Kuznetsova et al. reported graft copolymers of collagen and PAA using TBB as initiator. Graft copolymers were synthesized at three different temperatures. The resulting copolymers were characterized in terms of their structural morphology, tensile mechanical property, and cytotoxicity for potential use as scaffolds. This manuscript may be published after properly addressing the following comments.
1. In the introduction, some discussion regarding the purpose of preparing collagen based hybrid materials should be added.
2. Also in the introduction, discussion to clearly differentiate this study from previous studies on collagen-based graft copolymers should be included. In particular, as the authors already published studies on collagen-PMMA copolymers (ref 15), what is the motivation behind replacing PMMA with PAA?
3. Diagrams 1 and 2 are missing figure legends.
4. At higher polymerization temperatures (especially at 60C), the degree of AA conversion is lower and the copolymer MW is smaller. Can the authors provide some explanation to this observation?
5. For the graft copolymers synthesized, the ratio of PAA content is ~40%, but the increase in MW is very small (only ~10-20 kg/mol). This suggests that the PAA chains are very small. Can the authors provide some estimates on the size of PAA grafts? Can they be even considered as PAA polymers?
6. The authors stated that based on SEM images, the collagen fibers changed to gelatin form at higher temperature of 60C. It is not clear to me how that conclusion was made. Please discuss the SEM images in more detail to help the readers to see the structural differences between the fibers.
7. For the cytotoxicity test, cell morphologies on the different samples are described but no images are shown. Please include some representative cell images to support your description.
8. Also, regarding the cytotoxicity test, it is somewhat obvious that presence of AA monomer can lead to toxicity effects. The better comparison should be between PAA-g-collagen (60C) and PAA-g-collagen (25C), as the two have collagen present in different forms according to the authors.
Author Response
Dear, Reviewer! We have sent the answer in an attached file.

Reviewer 2 Report
Manuscript ID: polymers-1983031
Title: Graft polymerization of acrylamide in an aqueous dispersion of collagen in the presence of tributylborane
Dear Authors,
In this study, polyacrylamide (PAA) has been synthesized in the presence of acetic acid, fish collagen, tributylborane using graft polymerization technique. This work would give a good contribution to other polymer studies.
It is valueable to the related researchers and suggested to be published after minor revision by the following points:
· In the abstract part, the superior characteristic results of the synthesized material should be included and a striking summary should be made by specifying the application area.
· The sentence “The alkylborane-oxygen system is the most promising” in line 27 on page 1 should be revised by adding the answer of “for what purpose”.
· It would be appropriate to rewrite this sentence and design it in a more understandable way: “It should be noted that polymerization in the presence of an alkylborane- 54 oxygen initiator system requires no heating or UV irradiation, this being a particular ad- 55 vantage for the encapsulation of heat-sensitive hydrophilic active substances into silicone 56 copolymer particles [10, 11], in 3D printing [10], in the synthesis of hybrid products incor- 57 porating natural polymers [15–18] etc”.
Thus, the need for this article can be better emphasized.
· The order of the tables in the article needs to be changed. The table referred to in the materials and methods section given first should be Table 1.
· Similarly, it is not correct to start with Figure 3 since the Materials section comes first in the reading order. Figure numbering should be in the order given in the text in the article.
· It is also recommended to change the general editing in the manuscript. For example, instead of continuing the "Materials and Methods" section until 2.10, the synthesis parts can be given in Materials and Methods, the details of the devices used in the experimental study can be given in a sub-title called “Characterization”, and the others can be gathered under the main title of "Results and Discussions". Afterwards, integrity is ensured by giving "Conclusion".
· It would be appropriate to create an appropriate discussion section in the manuscript by comparing the findings in Table 5 with the literature.
· In “Results and Discussion”, the results found in the literature by different research groups should be compared with the results of the research by quantitatively (please, give exact numbers). In this way, the difference of the proposed manuscript from the literature should be revealed. The improvement achieved with the modification should be examined numerically. Related references must be added.
Thank you for your interest and best regards
Author Response

(The authors gave the same response as above.)

Reviewer 3 Report
The authors described a method using fish collagen, acrylamide, and tributylborane (TBB) to synthesize graft copolymers of collagen and polyacrylamide at low temperatures. They characterize the polymers by FT-IR, GPC, and SEM as well as the mechanical properties and cytotoxicity of the copolymers. However, the novelty of this material is not clearly stated despite the fact that the grafting reaction by TBB was already studied as well as the grafting reaction as they also stated in the introduction. Therefore, the reviewer suggests reconsidering this paper after the following questions are also addressed:
1. The arrow used in Diagram 2 was not clear. It has an arrow that seems to point from either C of the C=O bond, C-C bond of the C-C=O on the left-hand side, or another monomer. This is not clear. Also on the right-hand side, a single electron was moved from the oxygen to somewhere, which is also not clear. In the case of a single electron, a single barbed arrow should be used.
2. In figure 1, there are supposed to have 5 curves, however, it may be because the selection of the color is not obvious enough or one curve (#4) is totally missing from the plot. The representation of Fig 1 in terms of the arrangement, ticks’ selection, etc. makes it hard to compare the difference directly. The peak mentioned in the text should be labeled for a more direct visual comparison. e.g. 1640, 1540, 1720 cm-1. They are all between 1300-1800 cm-1.
3. Although this paper emphasizes the corporation of collagen, seems the mechanical properties of PAA/gelatin copolymer created at high temperatures are superior to PAA-g-collagen. The better performance of broken-down gelation should be specified in the introduction.
4. In terms of the breakdown of collagen at higher synthesis temperatures, are all collagens broken down or just partially? Is there any way to quantify this portion? This would be a significant point because in this paper what actually has better performance is the gelatin instead of the collagen, thus, if you solely change the content of gelatin in the collagen polymer, would this already be good enough to change the mechanical properties? In this case, what is the role of PAA?
Author Response

(The authors gave the same response as above.)

Round 2
Reviewer 1 Report
The authors provided sufficient responses to the comments. I have no further concerns regarding the publication of this manuscript.
Reviewer 3 Report
The authors have made the corresponding changes and revised the manuscript. Therefore, I suggest publishing this paper.